# Estimating the contribution of CD4 T cell subset proliferation and differentiation to HIV persistence

Daniel B. Reeves [1,2,13] ✉, Charline Bacchus-Souffan[3,13], Mark Fitch[4], Mohamed Abdel-Mohsen [5], Rebecca Hoh[6], Haelee Ahn[7], Mars Stone [8], Frederick Hecht [7], Jeffrey Martin [9], Steven G. Deeks [6], Marc K. Hellerstein [4], Joseph M. McCune[10], Joshua T. Schiffer [1,11,12,13] & Peter W. Hunt[7,13]

Persistence of HIV in people living with HIV (PWH) on suppressive anti-retroviral therapy (ART) has been linked to physiological mechanisms of CD4+ T cells. Here, in the same 37 male PWH on ART we measure longitudinal kinetics of HIV DNA and cell turnover rates in five CD4 cell subsets: naïve ($T_N$), stem-cell- ($T_{SCM}$), central- ($T_{CM}$), transitional- ($T_{TM}$), and effector-memory ($T_{EM}$). HIV decreases in $T_{TM}$ and $T_{EM}$ but not in less-differentiated subsets. Cell turnover is ~10 times faster than HIV clearance in memory subsets, implying that cellular proliferation consistently creates HIV DNA. The optimal mathematical model for these integrated data sets posits HIV DNA also passages between CD4 cell subsets via cellular differentiation. Estimates are heterogeneous, but in an average participant's year ~10 (in $T_N$ and $T_{SCM}$) and ~$10^4$ (in $T_{CM}$, $T_{TM}$, $T_{EM}$) proviruses are generated by proliferation while ~$10^3$ proviruses passage via cell differentiation (per million CD4). In simulations, therapies blocking proliferation and/or enhancing differentiation could reduce HIV DNA by 1-2 logs over 3 years. In summary, HIV exploits cellular proliferation and differentiation to persist during ART but clears faster in more proliferative/differentiated CD4 cell subsets and the same physiological mechanisms sustaining HIV might be temporarily modified to reduce it.

The persistence of chromosomally-integrated HIV DNA in CD4+ T cells is the primary barrier preventing people living with HIV (PWH) from achieving viral remission after stopping antiretroviral therapy (ART)[1,2]. HIV persistence has been associated to physiological mechanisms of CD4 cells[3,4] (e.g., homeostatic[5–7] and antigen-driven proliferation[8], cellular differentiation/maturation[9], and death). To help elucidate persistence mechanisms, it is critical to compare HIV DNA and CD4 cell dynamics as directly as possible.

To that end it is important to consider that integrated HIV DNA can be found in multiple CD4 cell subsets[6,9–11] (categorized by surface markers[12–14]) which have different physiological functions

(phenotypes) and maturational levels. For instance, at certain time points, higher proportions of HIV DNA have been found in more mature memory and effector CD4 cells, suggesting they are preferentially infected and/or expand HIV DNA through cellular proliferation[6,15–17]. On the other hand, longitudinally across individuals, HIV DNA appears to accumulate over time in less mature subsets that turn over less frequently[18]. However, no study to date has measured both CD4 cell turnover and HIV kinetics across subsets in the same individuals.

Mathematical modeling has continually proven useful to understand the kinetics and kinetic heterogeneity of HIV levels within a

---

person over time during suppressive ART[19–22]. In addition, modeling studies have sometimes inferred cellular rates using HIV as a molecular tag[23,24]. Our methodology builds upon a rigorous body of work using dynamical systems and population mixed effects modeling to quantitatively describe viral dynamics and recently, for multiple simultaneous data types[25–27].

Previously, most CD4 cell subsets have been shown to turn over several times per year in individuals without HIV[28,29]. These rates have been compared to HIV DNA decay rates (generally >4 year half-lives[30–32]), with the implication that HIV DNA in the reservoir must be replenished consistently while CD4 cells are born and die. Yet, a further complication is that genetically intact proviruses generally decay faster than defective ones[22,33,34], suggesting extrinsic factors like immune selection[35,36] may also influence viral persistence. Overall, the precise balance of processes that support reservoir maintenance remain incompletely characterized.

Here we measured cellular turnover in each of five resting CD4 cell subsets and changes in integrated HIV DNA levels within these subsets over 3 years in the same participants. We directly compared HIV DNA kinetics and cellular turnover rates within each subset and identified how these rates contribute to overall slow HIV DNA clearance. By selecting the most parsimonious mechanistic model for these combined data, we inferred the degree to which cellular proliferation and differentiation contribute to maintenance of integrated HIV DNA levels during suppressive ART. Finally, we simulated temporary modulations of proliferation and differentiation to highlight how minor changes in these processes might result in meaningful changes to HIV kinetics.

## Results

### Study cohort

The HOPE cohort consists of 37 PWH on suppressive ART (clinical and demographic information in Supplementary Table 1), 24 of whom underwent a 45-day deuterium labeling study to measure CD4+ T cell turnover rates and were reported previously[17] in a cross-sectional study. Here, we report a prospective 3-year longitudinal analysis of levels of integrated HIV DNA in distinct maturational CD4 cell subsets from all 37 HOPE participants and integrated these data with measured CD4 cell subset turnover rates. Follow up began 1–10 years after achieving viral suppression. Levels of integrated HIV DNA per million CD4+ T cells tended to be stable over time within individuals but differed between individuals by several orders of magnitude (Supplementary Fig. 1).

### Quantifying HIV DNA in CD4+ T cell subsets

From these longitudinal samples, resting (HLA-DR-) CD4+ T cells were isolated and sorted by flow cytometry into six CD4 cell subsets (sort schematic in Supplementary Fig. 2): naïve ($T_N$), stem-cell memory ($T_{SCM}$), central memory ($T_{CM}$), transitional memory ($T_{TM}$), effector memory ($T_{EM}$) cells, and a putative terminally differentiated ($T_{TD}$) population. As we observed contamination with $T_N$ in $T_{TD}$, the present analysis was focused on the first five sorted populations, each of which was sorted with high purity[17].

CD4+ T cell subset frequency was calculated as the ratio of subset cells per resting CD4 cells (Fig. 1A). $T_N$ and $T_{CM}$ were most common, each with a median across participants and time of ~25% of all resting CD4 cells. The infection frequency was then calculated as the number of integrated HIV DNA copies per million resting cells within each subset (Fig. 1B). Typically, ~1 in 1000 resting $T_{TM}$ and $T_{EM}$ harbored integrated HIV DNA, whereas the other subsets less commonly harbored HIV DNA[16]. Finally, by multiplying the subset frequency by the infection frequency, we derived the subset HIV DNA level which reflects the relative contribution of each subset to the measured HIV DNA, i.e., the number of integrated HIV DNA copies in a given subset per million total CD4 cells (Fig. 1C). Although not the highest in infection frequency, given its high subset frequency, $T_{CM}$ contributed the highest median HIV DNA levels, with ~100 infected $T_{CM}$ for every million CD4 cells. Median HIV DNA levels were generally lower but not significantly different in other memory phenotypes ($T_{TM}$ and $T_{EM}$). Considerable variability was noted within each subset and for each data type.

### HIV infected cells decay faster than non-infected cells in $T_{TM}$ and $T_{EM}$ (but not other) subsets

To determine if HIV DNA cleared differently in each subset, we used a statistical framework (log-linear mixed effects model) to assess changes in subset infection frequency over the 3-year study period (Fig. 2A). Although the decay rates were heterogeneous (and even positive, i.e., growing, in certain individuals), the average integrated DNA levels within $T_N$, $T_{SCM}$, and $T_{CM}$ did not significantly change over time (t test $p > 0.05$ against null hypothesis of no change), while those within $T_{TM}$ and $T_{EM}$ decayed slowly but significantly over time (t test $p < 1e-8$)

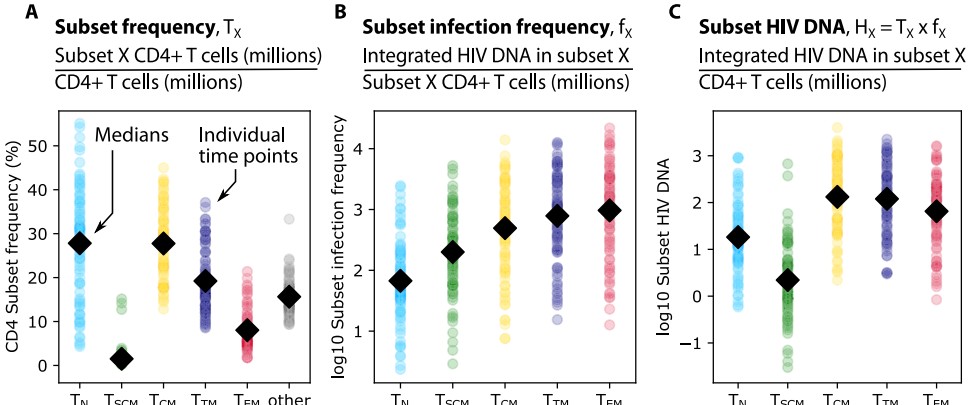

**Fig. 1 | Definitions and representation of study data.** From 37 PWH in the HOPE cohort, samples were taken at 1–3 time points over a 3-year period. Resting CD4+ T cells were sorted into five phenotypic subsets including naïve ($T_N$), stem-cell memory ($T_{SCM}$), central memory ($T_{CM}$), transitional memory ($T_{TM}$), and effector memory cells ($T_{EM}$). Three measurements were observed or calculated (panel headings): (**A**) subset frequency–the proportion of cells in each subset relative to total resting CD4 cells ("other" represents resting cells not among the five sorted subsets), (**B**) subset infection frequency–integrated HIV DNA in each subset per million subset cells, and (**C**) subset HIV DNA–the number of HIV DNA copies in a given subset per million CD4 cells. Colored dots indicate values from all participant time points and black diamonds represent means across all dots.

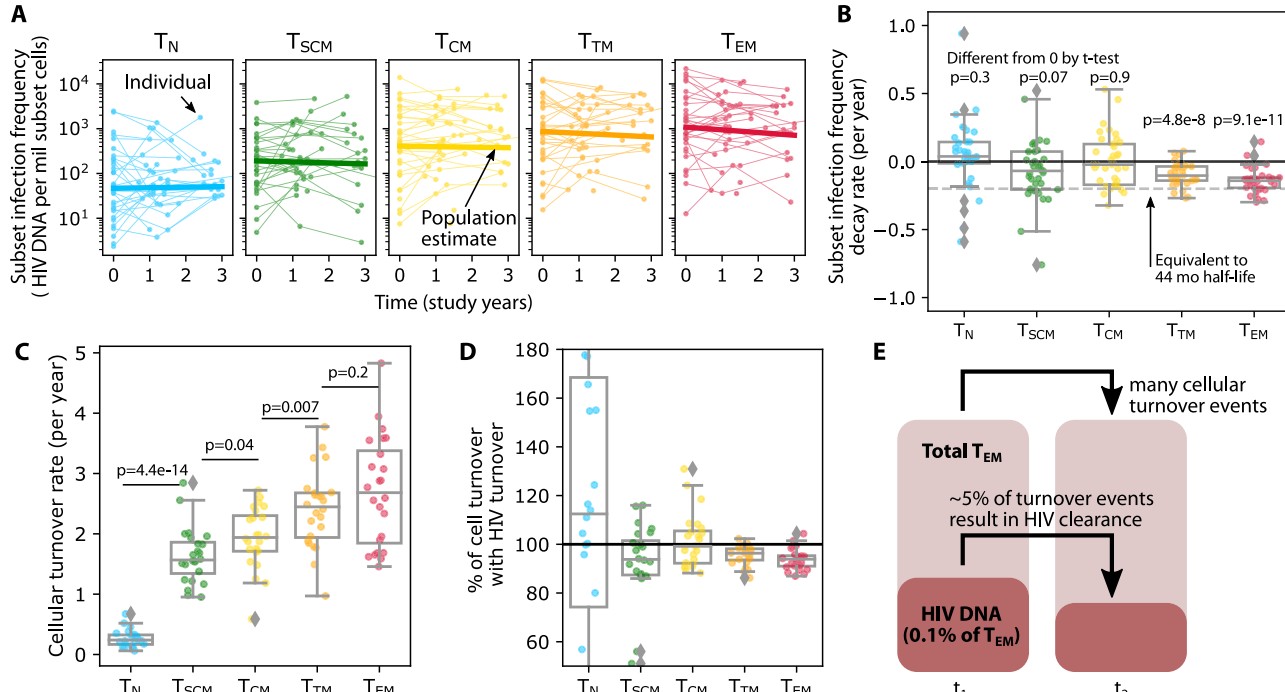

**Fig. 2 | The kinetics of subset HIV frequency vary by subset and are generally slower than cellular turnover. A** Longitudinal kinetics of HIV subset infection frequency in each cell subset: thin lines and dots are individual trajectories and thick solid lines represent the estimated average slopes from a log-linear mixed effects model. **B** Box plots of participants' decay rates−note that some are positive, meaning that HIV frequency increased. *P*-values indicate one-sided *t*-test against null hypothesis of no clearance. For scale, the decay rate equivalent to the QVOA reservoir benchmark 44 month half-life[30] is denoted with the dashed gray line. **C** Cellular turnover rates derived from deuterated water labeling in 24 of these 37 individuals. *P* values indicated paired two-sided *t*-tests with non-equal variance.

Magnitudes of cellular turnover rates (in non-$T_N$ subsets) are much higher than HIV decay rates−note difference in y-axis scales in (**C**) versus (**B**). **D** The % of cellular turnover that is accompanied by HIV turnover (Methods). Values close to 100% indicate that HIV is typically repopulated when cells turn over. In (**B–D**) box plots indicate median (center line), interquartile range (box), 1.5x interquartile range (whiskers), and outliers (gray diamonds). Each dot (*N* = 24) represents an individual. **E** Cartoon example for $T_{EM}$: in a year, there is frequent cellular turnover, which is infrequently (~5% of events) accompanied by elimination of HIV-infected cells, resulting in the observed slight decay of HIV DNA.

(Fig. 2B). Accordingly, $T_N$ and $T_{CM}$ rates were significantly different from $T_{TM}$ and $T_{EM}$ rates (pairwise *t* tests *p* < 0.005 according to Bonferroni correction for multiple comparisons). Estimated median half-lives were 81 and 59 months for $T_{TM}$ and $T_{EM}$, respectively. A declining subset infection frequency implies that HIV-infected cells decay faster than non-HIV-infected cells in that subset, suggesting an active process whereby HIV-infected cells are selectively removed.

**Measuring cellular turnover via deuterium labeling**
We used deuterated water labeling[37] performed on 24 of the 37 HOPE participants to estimate cellular turnover rates in each subset[17]. In these experiments, the cellular turnover rate is derived from modeling the proportion of cells that take up a deuterium label during a 45-day labeling period (model schematic in Supplementary Fig. 3). More specifically, the fraction of cells that divided during exposure to deuterated water is calculated[37–39]. Although what is initially measured from deuterium incorporation into genomic DNA is S-phase cell division, or proliferation[40], we instead use the term turnover rate here because this rate represents the combination of all mechanisms that impact levels of deuterium in a subset, including migration/trafficking and/or differentiation. For instance, labels in a given subset can rise due to maturation of a labeled progenitor cell or fall due to further maturation[41]. Cellular turnover rates ranged across subsets from slowest ($T_N$ median 0.2/year) to most rapid ($T_{EM}$ median 2.6/years) (Fig. 2C). Turnover rates were generally more rapid in more differentiated subsets, with the greatest differences between $T_N$ to $T_{SCM}$ and $T_{CM}$ to $T_{TM}$ (pairwise *t*-test *p*-values in Fig. 2C). A turnover rate of 1 per year corresponds to a half-life of 8.3 months, so these CD4 subsets have median half-lives of 35, 5.3, 4.3, 3.4, and

3.1 months, respectively. Considerable variability was noted within each subset.

**CD4+ T cell turnover is often but not always accompanied by HIV DNA turnover in certain subsets**
In all subsets except $T_N$, the cellular turnover rate was roughly an order of magnitude faster than the rate of decay of HIV-infected cells (compare Fig. 2B, C). This suggests that cellular turnover of HIV-infected cells does not usually result in removal of HIV DNA. We therefore estimated the percentage of cellular turnover events that might also be accompanied by HIV turnover rather than HIV clearance (Methods). For the five subsets respectively, we calculated medians of 112, 94, 99, 96, and 94% (Fig. 2D). In $T_N$, this number is greater than 100% suggesting some increases in HIV DNA in this subset; however, there was very high variability across participants making the median less reliable. Additionally, the much lower cellular turnover rates invoke lower signal compared to noise in the deuterium labeling measurements, potentially reducing precision. In the $T_{CM}$ subset, we estimate that cellular turnover almost always results in HIV turnover, so HIV DNA does not necessarily decline. Finally, in $T_{SCM}$, $T_{TM}$, and $T_{EM}$, 94−96% of cellular turnover can be associated with HIV turnover. That is, roughly 5% of cellular turnover events are accompanied by clearance of HIV DNA in these subsets (see example for $T_{EM}$ in Fig. 2E). Together, these results indicate that most, but not all, events that increase cell numbers−cellular proliferation and other mechanisms contributing to turnover−are accompanied by concomitant increases in HIV DNA. Any slight imbalance towards cell number increases without HIV increases could drive decay of HIV DNA in certain CD4 subsets.

## Mechanistic modeling of subset HIV DNA suggests differentiation rapidly passages HIV through CD4+ T cell subset maturation pathways

CD4+ T cell subsets are connected to one another by known steps of lineage maturation[14]. Previously, in this cohort, we found HIV DNA integrated into identical human chromosomal sites among $T_{CM}$ and $T_{TM}$ and $T_{TM}$ and $T_{EM}$ subsets, a strong sign that differentiation of HIV-infected cells can occur[17]. Moreover, HIV DNA frequencies and levels were found to correlate between certain subsets (Supplementary Fig. 4). Yet, the relative degree to which differentiation into a given CD4 cell subset versus proliferation within that subset contributes to HIV DNA persistence remains unclear. Therefore, we next sought to model HIV DNA levels with a mechanistic model that included specific rules of cellular proliferation, death, and differentiation.

We developed a variety of models inclusive of different mechanistic processes and degrees of complexity (Table 1, see Methods for equations and text describing assumptions). The list of models encodes scenarios in which HIV DNA levels are governed by one or more mechanisms including slow decay, proliferation, and cell differentiation between subsets. A schematic and table of definitions illustrates the rates we consider (Fig. 3A, B). We then tested these models for fit against levels of subset HIV DNA (e.g., Fig. 1C). Importantly, this is a different data type than in Fig. 2 and provides a common denominator of million CD4+ T cells for each subset. In our model, the levels of HIV DNA are linked across subsets, allowing proliferation and differentiation rates to be directly compared.

Models were ranked by their accuracy (fit to data) but also penalized for complexity using information criterion. The selected model (Fig. 3C, Supplementary Movie 1) ranked best by both Akaike and Bayesian information criteria[42] (AIC and BIC, Table 1). In this best model, each subset level of HIV DNA $H_s$ has a repopulation rate $\theta_s$ that encapsulates the balance of cell proliferation and death. Cellular differentiation passages HIV DNA between subsets $i$ to $j$ with rate $\phi_{i:j}$. Because we did not include the terminally differentiated subset ($T_{TD}$) due to $T_N$ experimental contamination, we could not estimate $T_{EM}$ clearance and differentiation rates simultaneously. Therefore, we explicitly note a combination of the two phenomena (see * in Fig. 3C). We also constrained parameter estimation to ensure rates for each subset were no larger than observed cellular turnover rates for that subset (Supplementary Fig. 5A, B). When this constraint on parameter space was relaxed, some models performed slightly better, but our initial best model remained second only to a model with the same structure but including biologically unrealistic rates (Table 1).

Therefore, for the remainder of the analysis, we proceeded with this more conservative model.

Qualitative features of model selection provide several mechanistic results. First, all models lacking differentiation had significantly poorer fit compared to the optimal model ($\Delta$AIC > 2, Table 1). A model that attempted to explain HIV levels through differentiation without cell proliferation was substantially worse than the optimal model ($\Delta$AIC = 85). The selected model includes passaging of HIV DNA along CD4 maturation pathways (i.e., linearly from least to most differentiated subsets) but additionally was improved by the addition of "skip" differentiation from $T_N$ to $T_{CM}$, and from $T_{CM}$ to $T_{EM}$. A simpler model with purely linear differentiation $T_N > T_{SCM} > T_{CM} > T_{TM} > T_{EM}$ was ranked 3rd but did not provide as strong a fit to data (Table 1). Together, these findings suggest differentiation is necessary but not sufficient to precisely describe HIV DNA dynamics in CD4 cell subsets over time.

To potentially broaden the applicability of this model, we provide a table of initial conditions, mean and standard deviation of population rates, and estimated variability of HIV DNA data (Supplementary Table 2).

### Sensitivity analysis on model selection

To assess whether the sparse 3-year sampling could have resulted in observations favoring a model with skip differentiation, we simulated the best-fit version of the model with linear differentiation, added appropriate noise, and sampled time points per the 3-year study scheme (Supplementary Fig. 6). We then refit this model to the linear- and skip-differentiation models. As expected, the linear differentiation model fit these data better than the skip-differentiation model ($\Delta$LL = 1.5, $\Delta$AIC = 10 compared to skip-differentiation model). This sensitivity analysis illustrates how model selection can be self-consistent, such that data generated with a given model contains enough information to recover the same model via model selection. In addition, it supported that the skip differentiation model was not innately favored based on noise or the sampling scheme.

### Estimating HIV DNA decay half-lives in the model inclusive of cellular differentiation

With some exceptions, model fits were excellent across highly variable subset trajectories (see 18 of 37 fits for participants with three time points, Fig. 4A). The overall population trends for each subset show that, notwithstanding some degree of heterogeneity, the average integrated HIV DNA level decays per million CD4+ T cells in 4/5 subsets

## Table 1 | Results of information theoretic mathematical model selection on integrated HIV DNA per million CD4+ T cells

| Rank | Model | ΔLL | N rates | ΔAIC | ΔBIC |
|---|---|---|---|---|---|
| 1 | Differentiation with skips: subsets can proliferate and die and are connected from least to most differentiated but additional connections are possible (e.g., $T_N > T_{CM}$.) | 0 | 11 | 0 | 0 |
| 2 | **Constrained differentiation with skips**: same as 1 but with limits on maximal differentiation rates (no greater than cell turnover) based on biological plausibility. | 10.5 | 11 | 10.5 | 10.4 |
| 3 | Linear differentiation: subsets can proliferate and die and are connected from least to most differentiated. | 25.4 | 9 | 17.4 | 10.9 |
| 4 | Carrying capacity: integrated HIV DNA in each subset is assumed to have an equilibrium value such that levels away from this value return through logistic growth/shrinking. | 28.7 | 10 | 24.7 | 21.4 |
| 5 | Linear differentiation linked to proliferation: a mathematical formulation in which some proportion of proliferation leads to differentiation. | 44.9 | 10 | 40.9 | 37.6 |
| 6 | No differentiation: subsets can only proliferate and die. | 84 | 5 | 60 | 40.6 |
| 7 | Constrained linear differentiation: same as #3 but with limits on maximal differentiation rates based on biological plausibility. | 75 | 9 | 67 | 60.5 |
| 8 | Carrying capacity 2: same as #4 with a different mathematical form for equilibration. | 73.2 | 10 | 69.2 | 65.9 |
| 9 | Only differentiation: subsets have no proliferation/death or net repopulation rates. | 113.1 | 4 | 85.1 | 62.5 |
| 10 | Forced clearance: repopulation rates must be negative, and no differentiation is included. | 136.4 | 5 | 112.4 | 93 |

Constrained differentiation with skips was chosen as the optimal model (see bolded rank 2) as best BIC given biologically realistic parameters. Δ denotes differences from the absolute best model (rank 1). N rates is included to indicate model complexity (more estimated rates is more complex).
*LL* log likelihood, *AIC* Akaike information criterion, *BIC* Bayesian information criterion.

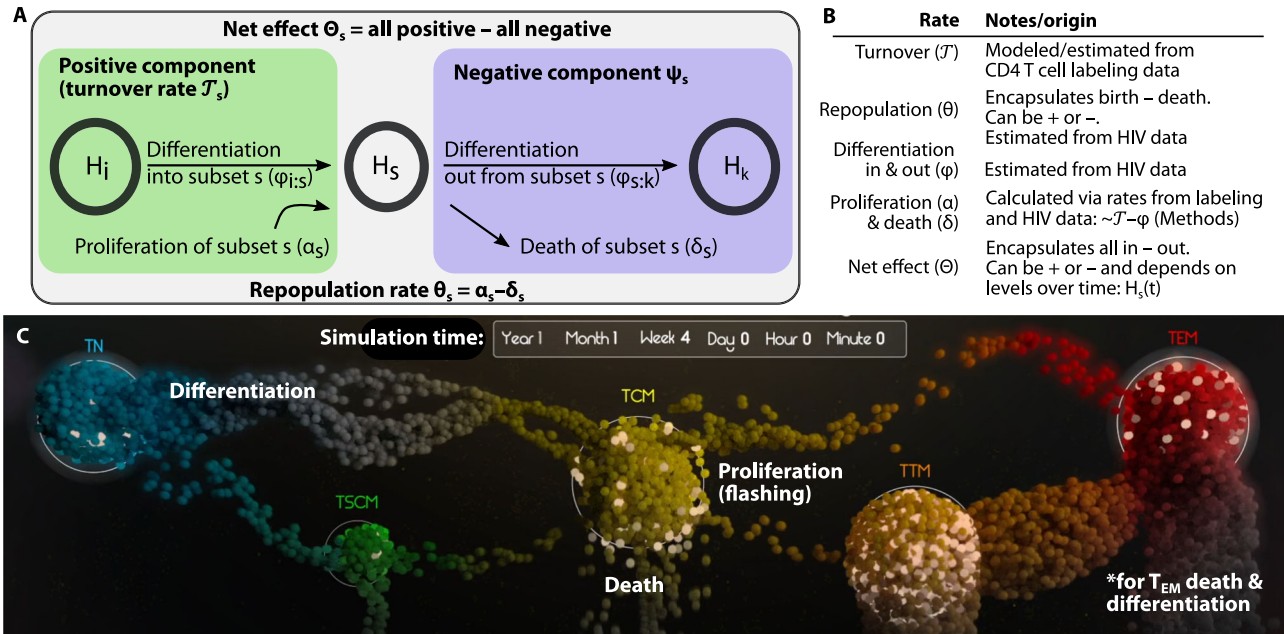

**Fig. 3 | Modeling subset HIV DNA dynamics via physiological mechanisms of T cells including proliferation, differentiation, and death.** Model schematic (**A**) and definitions (**B**) of model rates for a single subset. Net effect rates Θ describes the total kinetic rate summing all modeled mechanisms governing HIV DNA so can be positive or negative for each subset. The turnover rate represents the positive contribution to cellular turnover, estimated via the labeling study. Our mathematical model estimates the repopulation (θ) and differentiation (φ) rates in and out of each subset. Therefore, we can calculate the proliferation (α) and death (δ) rates for each subset from turnover and differentiation. **C** The most parsimonious model of all combined subset HIV DNA levels included infected cell proliferation (dots flashing), death (dots falling and fading), and differentiation between certain subsets (dots moving). This image is a screenshot of the Supplementary Movie 1 which visualizes the system over time. The differentiation pattern that was most parsimonious included a general flow from least to most mature subsets, but also some "skip" patterns, i.e., $T_N$-to-$T_{CM}$ and $T_{CM}$-to-$T_{EM}$. With no further measured subset past $T_{EM}$, death and differentiation out could not be distinguished for $T_{EM}$ so we combined the two phenomena (see *).

with a half-life of: 4.3 years in $T_N$, 2.6 years in $T_{SCM}$, 3.2 years in $T_{CM}$ and 3.7 years in $T_{EM}$ (Fig. 4B). At the same time, HIV DNA levels in $T_{TM}$ appeared to increase (which implies no half-life). When HIV DNA levels in all subsets were summed, the net half-life across all subsets was calculated to be 5.4 years. Although these data are not inclusive of all CD4 cell subsets capable of harboring HIV genomes, and individuals have different timeframes of ART (i.e., see trajectories in Supplementary Fig. 1), these half-life estimates are within ranges of previously-estimated HIV DNA decay[22,32,43].

**Quantifying the contribution of cell proliferation, death, and differentiation to integrated HIV DNA persistence**
To compare and contrast the mechanisms underlying HIV persistence in the best model, we next directly applied the cellular turnover data to estimate the absolute number of integrated HIV DNA copies (per million CD4+ T cells) that enter and leave each subset pool during a typical year due to proliferation, differentiation in and out, and death (Methods, Fig. 3A, B).

In a typical year in the average individual, we calculated (Methods) that 1–10 HIV DNA copies per million CD4 T cells are generated by proliferation of $T_N$ and $T_{SCM}$ while 100–1000 copies are generated by proliferation in $T_{CM}$, $T_{TM}$, and $T_{EM}$ (Fig. 5B). Meanwhile, similar numbers of HIV DNA copies are removed by death (Fig. 5D). These numbers imply that HIV DNA persists in a rapid and dynamic near-equilibrium state (Supplementary Movie 1). At the same time, few HIV DNA copies per million CD4+ T cells enter $T_N$ and $T_{SCM}$ (Fig. 5A), and 1–10 copies exit those subsets (Fig. 5C) due to differentiation. On average, ten copies enter, and 100 copies leave $T_{CM}$ due to differentiation (Fig. 5C). The unequal differentiation in and out then requires a slight imbalance favoring proliferation over death (Fig. 5B vs. Fig. 5D) to maintain $T_{CM}$ near equilibrium. $T_{TM}$ differentiation was almost balanced (mean -100

copies in, ~70 copies out in Fig. 5A vs. Fig. 5C). We could not distinguish $T_{EM}$ outward differentiation from death using these data since terminally differentiated cells were not studied in this analysis. Considerable variability was noted across participants within each subset.

Next, we compared mechanisms relative to one another by calculating the percentage of creation (differentiation in and proliferation) and removal (differentiation out and death) events from each mechanism and for each cell subset (Fig. 5E). Proliferation was the dominant mechanism contributing to the persistence of integrated HIV DNA in $T_N$, $T_{CM}$, and $T_{TM}$. However, differentiation inward may play an important role in maintaining HIV genomes in $T_{SCM}$ and $T_{EM}$. Differentiation outward was an important mechanism particularly for $T_N$ and $T_{CM}$, in which removal was projected to occur more through differentiation than death. $T_{EM}$ are known to proliferate frequently and had the highest cellular turnover rates. However, the absolute contribution of proliferation estimated here was lower than differentiation in. If HIV DNA dynamics mirror cellular dynamics measured with deuterated water experiments, this suggests that cellular turnover of HIV-infected $T_{EM}$ may particularly be influenced by differentiation.

In summary the model portrays typical HIV DNA levels as a rapidly proliferating, dying, and differentiating population that, in aggregate, maintains a nearly equilibrated system such that integrated HIV DNA only decays slowly and only in more mature CD4 subsets (Supplementary Movie 1). Importantly, proliferation remains the predominant mechanism in the generation of integrated HIV DNA. $T_N$ and $T_{SCM}$ contain less HIV DNA; therefore, the absolute HIV DNA creation and removal in those subsets is orders of magnitude smaller than that found in memory subsets. Proliferation is of particular impact in the context of $T_{CM}$ and $T_{TM}$: when coupled with differentiation outward (to one or more subsets), these subsets contribute meaningfully to HIV DNA persistence in the rapidly dying/differentiating $T_{EM}$ pool (Fig. 5F).

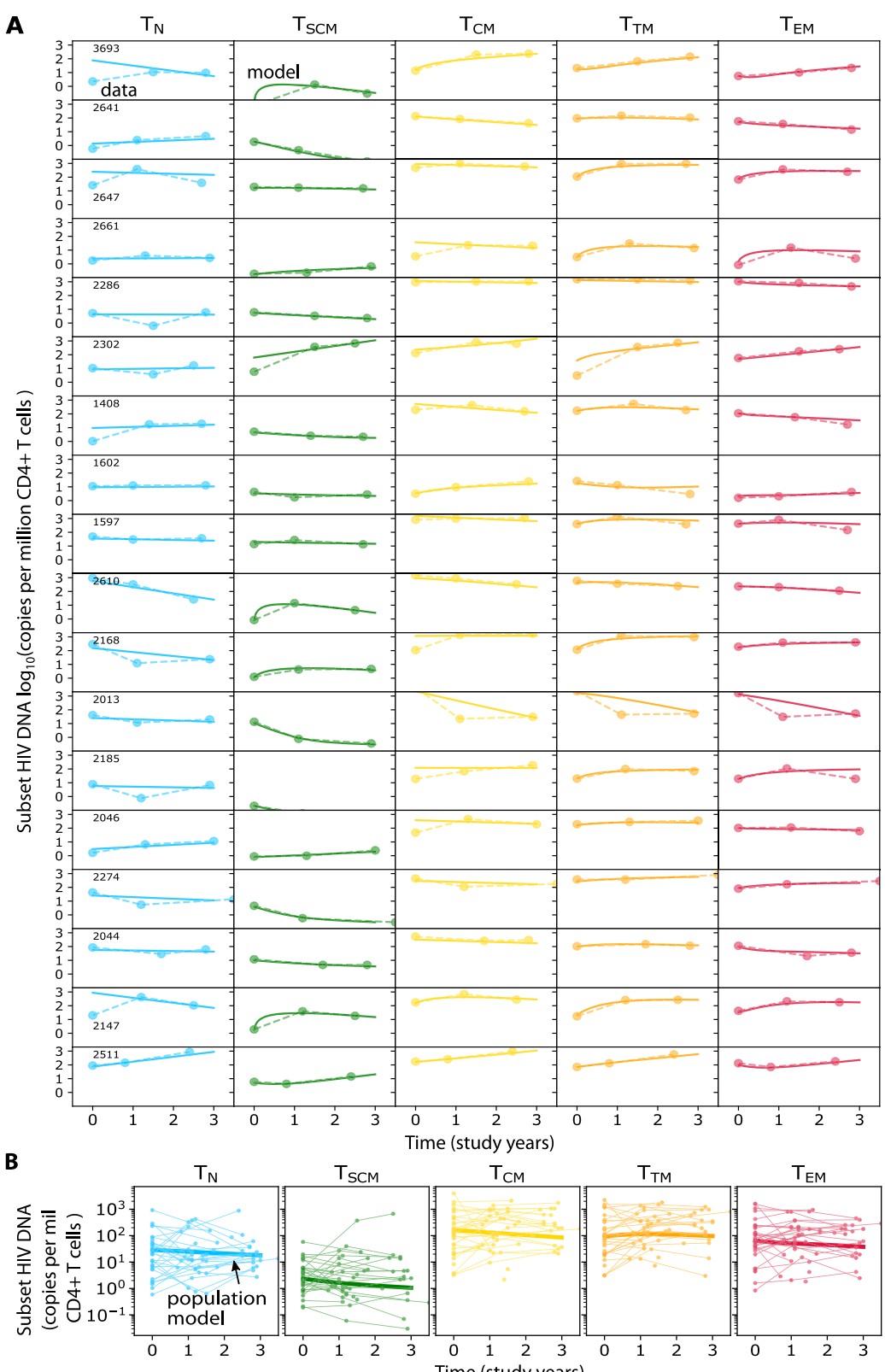

**Fig. 4 | Modeling including proliferation and differentiation recapitulates individual subset HIV DNA kinetics. A** Model fits (solid lines) of subset HIV DNA levels (dots/dashed lines) for all participants having 3 longitudinal measurements ($N = 18$). **B** Population model (solid lines) estimates of subset HIV DNA (copies per million CD4 T cells) to all longitudinal participant data (dots with thin lines).

### Modeling cell-associated HIV RNA

We also fit models with no differentiation, linear differentiation, and our favored model with skip differentiation to cell-associated HIV RNA (caRNA) levels measured in the same participants (Supplementary Fig. 7). For these data, the model without differentiation was optimal via AIC. In line with observations for HIV DNA, caRNA levels per million CD4 T cells appeared to increase slightly in $T_{CM}$ and decrease in $T_{EM}$ in these participants. But unlike for DNA,

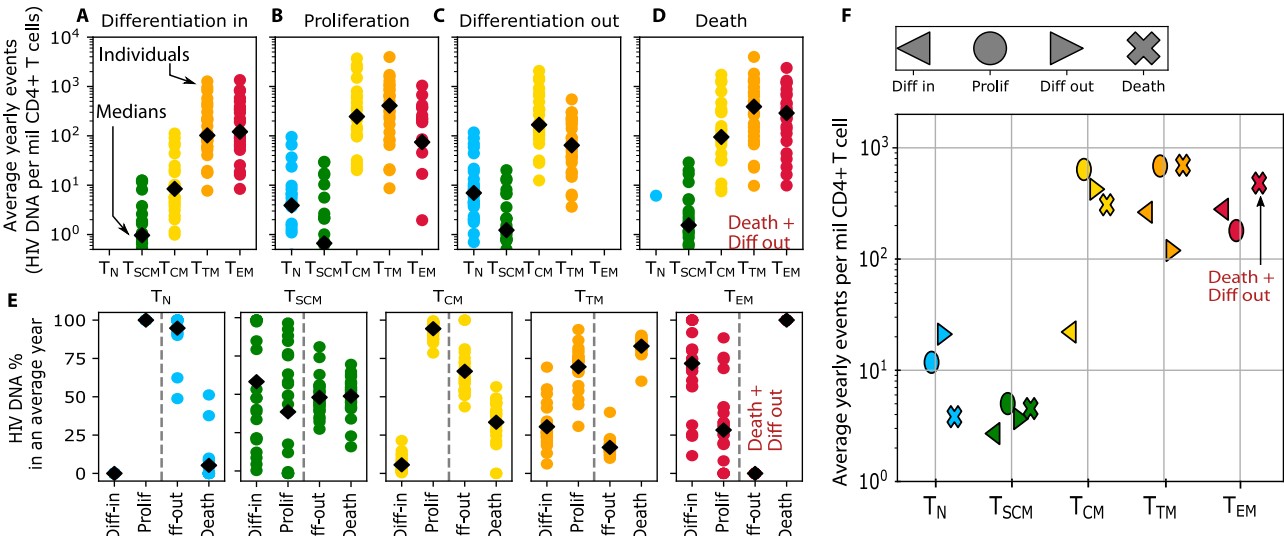

**Fig. 5 | Absolute and relative contribution to HIV reservoirs by cell proliferation, death, and differentiation. A–D** Absolute contributions to HIV subset DNA by differentiation in, proliferation, differentiation out, and death of each subset. **E** Relative contribution of each mechanism to each subset. Positive (persistence) and negative (clearance) contributions are treated separately for % calculations. Differentiation out and death of $T_{EM}$ are grouped together because the lack of terminally differentiated cells in this analysis precluded identification of both rates. In **A–E**, estimate for each individual ($N = 24$) are shown as colored dots and black diamonds indicate means across individuals. **F** The absolute contribution of each mechanism averaged across all individuals.

RNA increased in $T_{TM}$. Together, these data suggest that RNA levels are less tightly connected across subsets, potentially because RNA is generated by DNA and additional variability in this process reduces correlations.

## In silico knockout demonstrates the theoretical capacity of reservoir reduction through reduced cell proliferation and/or enhanced cell differentiation

Mechanistic modeling provides the valuable ability to project the dynamics of HIV DNA persistence in the context of perturbed CD4+ T cell subset proliferation and/or differentiation. Thus, we used the model to simulate three therapeutic scenarios over a period of three years: ART alone (Fig. 6A), ART with anti-proliferative therapy that reduces cellular proliferation for all subsets by a factor of 2 (Fig. 6B), and ART with enhanced differentiation therapy that increases differentiation for all subsets by a factor of 2 (Fig. 6C). We calculated changes in HIV DNA per million CD4+ T cells over time. For ART alone, (as observed in the raw experimental data) we projected a relatively minimal median change and wide variability inclusive of increases and decreases in all subsets. For ART and anti-proliferative therapy, median HIV DNA across subsets dropped by 300 copies (or ~90%) with most simulations resulting in overall decrease. For ART and enhanced differentiation therapy, median HIV DNA across subsets dropped by 200-300 copies (or ~80–90%) with slightly more simulations inclusive of no change or increase versus anti-proliferative therapy.

## Discussion

Here, we addressed the mechanistic basis for HIV persistence during ART in different phenotypic subsets of CD4 + T cells. We measured both longitudinal levels of integrated HIV DNA and cellular turnover rates in five resting CD4 cell subsets in ART-suppressed people living with HIV (PWH). In agreement with previous studies in adults and children[6,15,17,44], HIV DNA in these individuals was most commonly found in central, transitional, and effector memory subsets ($T_{CM}$, $T_{TM}$, and $T_{EM}$). Although total levels of naïve CD4 T cells ($T_N$) were as high, if not higher than those with a memory phenotype, $T_N$ were much less frequently found to harbor integrated HIV DNA, consistent with observations that memory subsets are easier to infect[45,46] and/or that HIV DNA accumulates more quickly within them[7].

We documented that HIV decays more rapidly in differentiated CD4 cell subsets ($T_{TM}$ and $T_{EM}$) vs. less mature subsets ($T_{CM}$ and $T_N$). This explains why HIV DNA appears to accumulate in less-differentiated subsets, as observed in a prior cross-sectional study[17]. It is possible that proliferation and/or differentiation in these subsets promotes HIV expression and immune recognition[47], leading to preferential removal of latently infected cells[48]. However, $T_{CM}$ also commonly proliferated, so more experiments are needed to refine mechanisms in each subset.

Deuterium labeling data from these PWH demonstrated that turnover rates of predominantly uninfected memory CD4 cells were approximately tenfold faster than HIV DNA decay rates. Therefore, we concluded that HIV-infected cells must frequently die and repopulate by cellular proliferation (and/or differentiation). Additionally, $T_N$ turned over substantially less frequently, such that in this subset cellular longevity of latently infected cells is a potential mechanism of reservoir persistence. Most importantly, HIV-infected cells must be slightly balanced towards death during cellular turnover to allow for the HIV DNA decay we observed in the most differentiated subsets.

Cellular differentiation naturally occurs in the context of homeostasis of the total CD4+ T cell population[13,14,49]. However, the contribution of CD4 cell differentiation to HIV persistence has mostly been discerned indirectly[9] and the magnitude of differentiation, especially as compared to cellular proliferation, has not been quantified. We also observed strong associations between HIV levels in different cell subsets over time. Therefore, we tested mathematical models of HIV DNA levels that directly linked subsets and found that models inclusive of differentiation allowed for the best agreement with the data, strengthening the evidence that integrated HIV DNA is passaged from one subset to another through physiologic pathways of CD4 T cellular differentiation.

Our optimal model included "skips" in which HIV DNA was passaged from $T_N$ to $T_{CM}$ and $T_{CM}$ to $T_{EM}$ without going through intermediate $T_{SCM}$ and $T_{TM}$ subsets. There may be a mechanistic explanation for why apparent "skipping" is a better fit than linear differentiation. Indeed, it is hard to reconcile the speed of antigenic

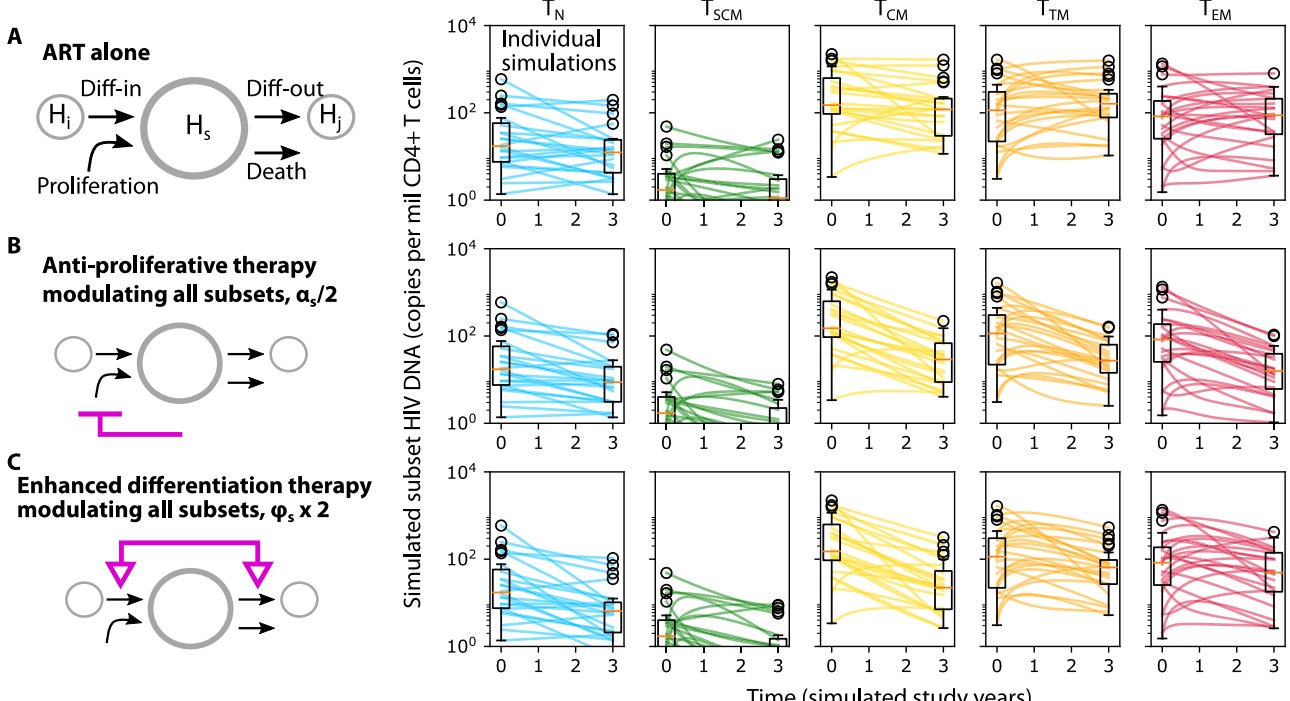

**Fig. 6 | Simulations of modulated HIV persistence mechanisms.** Projections of subset HIV DNA levels in all resting CD4+ T cell subsets during three theoretical therapeutic interventions: **A** ART alone, (**B**) ART and anti-proliferative therapy: 2-fold reduction in cell proliferation in all subsets, and (**C**) ART and enhanced differentiation therapy: 2-fold increase in cell differentiation in and out of all subsets. Box plots indicate median (center line), interquartile range (box), 1.5x interquartile range (whiskers), and outliers (open circles). Each line (*N* = 24) represents a simulation using parameters from each individual.

response with a model in which $T_{CM}$ must become $T_{TM}$ (with 3 months half-lives) before becoming $T_{EM}$. As different viral infections are controlled by phenotypically different CD8+ T cell subsets[50], it may be that certain CD4 cell subsets respond to different antigens.

On an absolute scale, cellular proliferation was confirmed to be the dominant mechanism of reservoir persistence[51], accounting (using the best model) for 10–10,000s of new HIV DNA copies per million CD4 cells in a given year. The upper end of this range is comparable to estimates of total reservoir sizes such that, in 1 year, individual cells carrying integrated HIV DNA may be completely refreshed while HIV DNA levels remain nearly constant. In addition to proliferation, the persistence of HIV DNA was also found to be driven by cell differentiation. An implication is that differentiation does not necessarily re-activate HIV expression and result in immune recognition. The overall picture from the model is one in which all subsets proliferate (and $T_{CM}$ in particular proliferate and differentiate rapidly), creating HIV DNA and passaging it onto more mature progeny.

Though we did not study HIV provirus clonality, the best model helps to mechanistically explain past observations of clonal HIV proviruses detected in different CD4 cell subsets[9–11,52]. Other recent data showed predominantly unique HIV sequences isolated from $T_N$, whereas those retrieved from $T_{EM}$ were mainly clonal[53,54]. Previously, we reported on HIV clonality in some HOPE study participants[17]. Oligoclonality was generally higher in more mature subsets and these subsets also had the highest degree of sharing of the same clonotypes. The present modeling provides some mechanistic insight for these observations: we estimate that much of the integrated HIV DNA found in $T_{EM}$ and $T_{TM}$ was likely generated upon cell proliferation and/or passaged onward from the highly proliferative (and therefore highly clonal) progenitor $T_{CM}$ subset. Thus, these subsets are both highly likely (in absolute terms) to be clonal and to share clones in common. However, while $T_N$ is still predominantly sustained by proliferation, their lower proliferation rates mean $T_N$ is relatively less clonal than other subsets.

Armed with the mechanistic model[55], we simulated in silico therapeutics and found that continually reducing cellular proliferation (anti-proliferative therapy[7]) or enhancing differentiation (akin to "rinse and replace"[56]) during suppressive ART could substantially reduce HIV DNA levels relative to the use of ART alone. These approaches achieve reduction in HIV DNA differently. It is assumed that the natural (slow) HIV DNA clearance rate in each subset arises from a balance of cell proliferation and death. Anti-proliferative therapies imbalances each subset individually, and HIV DNA clearance is projected to be faster in subsets with higher natural death (turnover) rates. Alternatively, enhancing differentiation does not increase clearance in each subset but rather pushes HIV DNA into the most differentiated compartments, in which HIV DNA clears more rapidly.

In all our simulations, sustained therapy for several years was required to meaningfully reduce HIV DNA, which presents both clinical and experimental challenges for validation. Nevertheless, a human study on IL-15 superagonist N-803, an anti-cancer drug that might promote differentiation[57], achieved a small but significant reduction in inducible HIV proviruses[58]. Individuals taking Dasatinib, a different anti-cancer agent that restricts antigen-driven and homeostatic proliferation of CD4+ T cells in PWH[59], also appeared to have lower HIV DNA levels than those taking ART alone, but whether this effect is driven by anti-proliferation requires more research[60].

Predictions about anti-proliferative or pro-differentiation therapy from the in silico models should be interpreted carefully. For instance, larger HIV clones found during ART were observed to be less likely to reactivate when ART was stopped (with a continuous relationship between probability and size[61]), perhaps because they are either genetically defective[62–65] or integrated within epigenetically silenced locations (graveyards)[35,36,66]. This could suggest that more proliferative clonotypes, which in turn might be more affected by anti-proliferative therapy, may be less relevant for predicting viral rebound. Because we did not have viral rebound data, we did not explore models that

included viral reactivation[67–69] (also precluding simulation of latency reversal agents[70]).

There are also experimental caveats to this work. CD4 cell subset categorization is inherently imperfect because identifying cells by cell surface markers requires defining thresholds and dichotomizing what is likely a continuum of cell maturation states[71]. In particular, $T_N$ may be heterogeneous to the point of resembling other phenotypes[72]. We could not distinguish the loss of HIV genomes in $T_{EM}$ through cell death or migration or differentiation outward because we did not successfully sort high-purity terminally differentiated cells.

On the modeling side, our absolutely best scored model admitted rates that were not necessarily biologically plausible. By constraining these rates, we derived a reasonable model that still fits data accurately. Going forward, it would be ideal to collect more temporally resolved data to refine these rates. Other simplifications include that we did not model HIV DNA influx into $T_N$ cells -- although small numbers of recent thymic emigrants and/or bone marrow progenitors can be infected[73,74]. Cellular trafficking to other anatomic compartments and the role of resident memory CD4+ T cells[75] were not explicitly modeled; on the other hand, the composite movement of cells in and out of tissues are likely balanced over the multi-year study time-scales in our study. Finally, on a conceptual level, cell differentiation and proliferation are fundamentally single cell/lineage properties, whereas we interpreted estimated rates as frequencies of cellular processes averaged over cell populations, which inherently minimizes within-host stochastic effects.

A strength of this study is the direct comparison of CD4 cell turnover and HIV DNA decay in the same participants and subsets. Still, most CD4 cells during ART are not HIV-infected, so it is unclear whether measured turnover rates precisely represent those of HIV-infected cells. HIV-infected cells that persist may be particularly biased toward cell survival and/or proliferation[76,77], or more likely to express signatures indicating resistance to immune-mediated killing[35,36,66]. Our modeling does not reach this level of genetic precision, but our observations of proportional DNA decay in more differentiated CD4 cell subsets ($T_{TM}$ and $T_{EM}$) indicate that survival mechanisms are likely insufficient to overcome clearance mechanisms in these subsets.

Finally, it would be desirable to estimate mechanistic contributions specifically to the persistence of intact proviruses, which are much more rare but known to clear more rapidly than defective HIV proviruses in the first years of ART[33,34,62,65,78]. Depth remains a challenge in many HIV reservoir studies, and filtering HIV DNA into both subsets and by intactness has admitted very low proviral counts[16]. We hope this limitation can be overcome in the future.

In summary, by examining HIV DNA levels and cellular turnover in CD4+ T cell subsets, we found that HIV DNA decays faster in differentiated CD4 cell subsets and quantified how both cellular proliferation and differentiation contribute to HIV persistence. Our simulations suggest that the same mechanisms that HIV exploits for its persistence might also be leveraged for its elimination.

## Methods

### Inclusion and ethics

All participants were over 18 years old and provided written informed consent for inclusion before they participated in the study. The study (NCT00187512) is an observational, prospective study of HIV-1 infected volunteers designed to provide a specimen bank of samples with carefully characterized clinical data. The study was conducted in accordance with the Declaration of Helsinki, and the protocol was approved by the University of California San Francisco Committee on Human Research.

### Study participant characteristics

Thirty-seven persons living with HIV (PWH) on ART were recruited between 2015 and 2019 from the clinic-based SCOPE and OPTIONS cohorts at Zuckerberg San Francisco General Hospital. Study participants returned yearly for 1-3 time points of follow up. The SCOPE cohort enrolls PWH with chronic HIV, whereas the OPTIONS cohort enrolls PWH < 12 months (before 2003) and <6 months (after 2003) following HIV antibody seroconversion. Viral suppression by ART was a requirement for study entry. Duration of viral suppression was estimated based on clinic records (typically assessed every 3–6 months). HIV acquisition timing for each participant was estimated as previously described[79].

### Isolation of CD4+ T cell subpopulations

All participants underwent leukaphereses performed as outpatients. PBMC were isolated and viably cryopreserved. Frozen PBMC were thawed and CD4 T cells enriched with the EasySep™ Human CD4 + T Cell Negative Selection Enrichment Kit (Stemcell). Cells were stained with Live/Dead Fixable Aqua (Life Technologies) and the following monoclonal antibodies cocktail: anti-CD3-FITC, anti-CD4-Alexa-Fluor700, anti-CCR7-PE-Cyanine7, anti-CD27-APC, anti-HLA-DR-APC H7, anti-CD57-Brilliant Violet 421, and anti-CD95-PE (Becton Dickinson) as well as anti-CD45RA-ECD (Beckman Coulter). HLA-DR- CD4 + T cell subpopulations were sorted on a FACS ARIA II flow cytometer (BD Biosciences) at >97% purity. Dry pellets were snap-frozen at −80 °C. Flow cytometry data were analyzed on FACSDiva v8.0.1 (BD Biosciences) and FlowJo v8.7 (Tree Star). Sorting schema is provided in Supplementary Fig. 2.

### Integrated HIV DNA quantification

Total DNA was extracted using the Allprep DNA/RNA/miRNA Universal Kit (Qiagen). Integrated HIV DNA copies were quantified with a two-step PCR reaction[80] using isolated genomic DNA for PCR amplification instead of whole cell lysates. Integrated HIV DNA was pre-amplified with two Alu primers and a primer specific for the HIV LTR region, in addition to primers specific for the CD3 gene to determine cell counts. Nested qPCR was then used to amplify HIV and CD3 sequences from the first round of amplification. Specimens were assayed with up to 500 ng cellular DNA in triplicate and copy number was determined by extrapolation against a 5-point standard curve (3–30,000 copies), using extracted DNA from ACH-2 cells.

### Cell-associated HIV RNA quantification

Total RNA was extracted using the Allprep DNA/RNA/miRNA Universal Kit (Qiagen) with on-column DNase treatment (Qiagen RNase-Free DNase Set). HIV RNA levels were quantified with a qPCR TaqMan assay using LTR-specific primers F522-43 (5′ GCC TCA ATA AAG CTT GCC TTG A 3′; HXB2 522-543) and R626-43 (5′ GGG CGC CAC TGC TAG AGA 3′; 626-643) coupled with a FAM-BQ probe (5′ CCA GAG TCA CAC AAC AGA CGG GCA CA 3′) on a StepOne Plus Real-time PCR System (Applied Biosystems, Inc.)[81]. Up to 500 ng of total RNA per sample were characterized in triplicate, and copy numbers were determined by extrapolation against a 7-point standard curve (1–10,000 copies). The input cell number in each PCR well was estimated using independent qPCR measurement of the cellular housekeeping human RPLP0 gene.

### Estimating the slope of subset infection fraction

To estimate the slope of HIV subset infection frequency (per million cells of each resting subset), we assumed that the longitudinal kinetics of each subset infection frequency $f_X$ followed an independent exponential model:

$$\dot{f_X} = \Delta_X f_X \qquad (1)$$

So that each subset (denoted by $X$) has a rate of change per year (or log-linear slope) $\Delta_X$. Using MONOLIX[25], we estimated the five values of $\Delta_X$. Importantly, we did not assume this rate was negative, such that increases (rather than clearance) were possible. Then, for subsets with

negative values of this rate, the half-life in years could be estimated as $hl = -\ln(2)/\Delta_X$.

## Calculating the percentage of cellular turnover events that result in HIV repopulation

In Fig. 2D, we used each subset infection frequency decay rate $\Delta_X$ and its matching cellular turnover rate $\mathcal{T}_X$ to calculate the percentage of cellular turnover events resulting in HIV repopulation. Assuming the net decay can be accounted for as a balance of turnover and repopulation $\Delta_X = r_X - \mathcal{T}_X$, the repopulation percentage is $r_X/\Delta_X$ or:

$$\frac{r_X}{\Delta_X} = \frac{\Delta_X + \mathcal{T}_X}{\Delta_X} \qquad (2)$$

## Normalized correlations between subset levels

Further evidence for connections between subsets emerged from a correlation analysis (Supplementary Fig. 4). For both subset frequencies and subset HIV DNA data, values were normalized to each individual's longitudinal average value (i.e., $\widetilde{f}_X(t) = f_X(t)/\langle f_X \rangle_t$). This procedure prevents spurious correlations (Simpson's paradox) related to large or small absolute reservoir sizes. Then, pairwise Spearman correlations were computed using the *SciPy* Stats package.

## Mechanistic mathematical models for subset HIV DNA

Our general model of the connected system of HIV DNA in each subset is governed by a system of differential equations that splits the kinetics of HIV DNA into the processes of proliferation, death, and differentiation between subsets. Others have used similar equations[82]. Each model can be written in vector form as:

$$\dot{H}_s = F(H_s | \theta_s, \phi_{k:s}, \phi_{s:k}) \qquad (3)$$

Where subset HIV DNA in each subset is the vector $H_s = \{H_N, H_S, H_C, H_T, H_E\}$, and the clearance and differentiation rates are written with the vectors as $\theta_s$ and $\phi_{i:j}$, respectively. Differentiation could be generally from different compartments into others so it is not necessarily the same sized vector in each model. The models tested are numbered as follows:

*Model 1* assumes each subset is independent and decays or grows independently (similar to the model used for subset infection frequency in Eq.(1)):

$$\dot{H}_s = \theta_s H_s \qquad (4)$$

Based on past observations of a net decrease in HIV DNA over years of ART, *Model 2* tested the hypothesis that all subset HIV DNA decays independently by using the same structure as Model 1 but forcing $\psi_s < 0$.

*Model 3* assumes a linear differentiation model whereby each subset had a decay term and differentiation terms in and out from most proximal subsets. There are, therefore, four differentiation terms: $\boldsymbol{\phi} = \{\phi_{N:S}, \phi_{S:C}, \phi_{C:T}, \phi_{T:E}\}$.

$$\dot{H}_s = \theta_s H_s + \phi_{i:s} H_s - \phi_{s:j} H_s \qquad (5)$$

*Model 4* assumed a more complex differentiation pattern derived from the significant correlations between subsets observed in Supplementary Fig. 4. In this model, there are 6 differentiation terms, the same four linear differentiation rates as in *Model 3*, and two additional skip terms: $\phi_{N:C}$ and $\phi_{C:E}$.

For models including differentiation, we generally assumed that the differentiation rate of HIV DNA into naïve cells from some unknown/unobserved compartment was zero: $\phi_{?:N} = 0$. This assumption is based on TREC content observations suggesting thymic

emigrants are not carrying HIV DNA frequently, if at all[17]. We also assumed differentiation out from $T_{EM}$ was zero: $\phi_{E:?} = 0$. There may be other terminally differentiated cells that $T_{EM}$ can transition into, but these were not observed in the study. Therefore, the clearance rate of $T_{EM}$ effectively covers death and differentiation out and is denoted $\psi_E$ rather than $\theta_E$ to make this explicit in Fig. 3.

As another approach, *Model 5* assumes each subset was independent and followed a logistic growth term with a carrying capacity. This tests the hypothesis that decay was not occurring and that HIV DNA levels in each subset had a rough equilibrium:

$$\dot{H}_s = r_s H_s (1 - H_s/K_s) \qquad (6)$$

Yet another approach (*Model 6*) more explicitly tested the hypothesis that proliferation and differentiation were linked. We assumed that some fraction $\zeta \in [0,1]$ of repopulation events are associated with differentiation:

$$\dot{H}_s = \theta_s H_s (1 - \zeta_{s:s+1}) + \theta_{s-1} H_{s-1} (1 - \zeta_{s-1:s}) \qquad (7)$$

As a final note, multiphasic decay is well documented for HIV DNA clearance after initiation of ART[32,83]. However, these phases are generally equilibrated within a year or two of ART initiation, which was irrelevant to our data.

## Model fitting and selection with population non-linear mixed effects modeling (pNLME)

Model fit and selection was performed using MONOLIX[25] software, which employs a population nonlinear mixed-effects (pNLME) approach. We assumed assay variability (noise) was log normal. Repopulation parameters were generally assumed to be normally distributed (allowing for negative values), and differentiation rates were generally assumed to be lognormally distributed. Population parameters were found to be uncorrelated, but across individuals, certain parameters were strongly correlated (Supplementary Fig. 5C). This finding suggests that those with higher rates in one subset tend to also have higher rates in others. Individual best fit parameters for each participant using the optimal model are collected in Supplementary Data 1.

## Imputing turnover rate to define mechanistic components

The underlying assumption of the repopulation rate is that it is a balance of proliferation and death, $\theta_s = \alpha_s - \delta_s$. To estimate these component rates, we use the cellular turnover data (Fig. 3B). We begin with a general equation for the $i$-th HIV DNA subset level from the best model and use a quasistatic assumption $\dot{H}_i = 0$ to indicate that cellular turnover mitigates a balance inward and outward of each subset. Yet, this balance has an absolute value $\mathcal{T}_i$ such that in some subsets, although net zero change occurs, there is more inward and outward flow. We therefore set inward and outward mechanisms (from Eq. (5)) equal and split repopulation into proliferation and death, leaving:

$$\Sigma_j \phi_{j:i}^{in} H_j + \alpha_i H_i = (\Sigma_k \phi_{i:k}^{out} + \delta_i) H_i. \qquad (8)$$

The turnover rate $\mathcal{T}_i$ (per year) then can be factored out of the rhs as $\mathcal{T}_i = \Sigma_k \phi_{i:k}^{out} + \delta_i$, such that the death rate can be defined as the turnover rate minus the differentiation rate out:

$$\delta_i = \mathcal{T}_i - \Sigma_k \phi_{i:k}^{out} \qquad (9)$$

And similarly solving $\Sigma_j \phi_{j:i}^{in} T_j + \alpha_i T_i = \mathcal{T}_i T_i$, leads to:

$$\alpha_i = \mathcal{T}_i - \Sigma_j \phi_{j:i}^{in} T_j / T_i \qquad (10)$$

Which we approximate by using the values of $T_i(0)$.

**Tracking equations to distinguish mechanistic contributions**

After imputing the turnover rates to define the mechanistic compartments, the HIV DNA created at any time $t$ (instantaneously in the interval $\Delta t$) due to each mechanism was computed. This computation occurs independently after solving the differential equations. Thus, the proliferation and death terms follow

$$H_s^{\text{pro}}(t) = \alpha_s H_s(t) \Delta t, \, H_s^{\text{death}}(t) = \delta_s H_s(t) \Delta t, \quad (11)$$

While the differentiation terms follow:

$$H_s^{\text{diff}-\text{in}}(t) = \sum_k \phi_{ks} H_k(t) \Delta t, \, H_s^{\text{diff}-\text{out}}(t) = \sum_k \phi_{sk} H_s(t) \Delta t. \quad (12)$$

## Reporting summary

Further information on research design is available in the Nature Portfolio Reporting Summary linked to this article.

## Data availability

Data used to generate figures is available at https://github.com/dbrvs/HOPE-modeling and in Supplementary Data 1.

## Code availability

All model code is freely available at https://github.com/dbrvs/HOPE-modeling. Python and in particular the Seaborn package were used to generate all figures.

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

## Acknowledgements

The authors sincerely thank each one of the HOPE study participants who devoted their time and bodies to our research and the clinical staff from the SCOPE and OPTIONS cohorts who made this project possible: Monika Deswal, Montha Pao, Heather Hartig, Marian Kerbleski, and Viva Tai. We thank Alex Carvidi, Elizabeth Sinclair, and Jeff Milush from the Core Immunology Lab for their assistance on this study; Mike Busch as well as the "Reservoir Assay Validation and Evaluation Network" (RAVEN) study group for their contribution to the collection of leukaphereses; and the DARE U19 Consortium for their support. This study was supported by the following NIH grants: K25 AI155224 (D.B.R.), R01 AI116368 and K24 AI145806 (P.W.H.), UM1 AI126611 (S.G.D.), and P30 AI027763. D.B.R. is extremely grateful for the support of the University of Washington/Fred Hutch Center for AIDS Research (CFAR, P30 AI027757) through a New Investigator Award that initiated this project.

## Author contributions

D.B.R., C.B.S., P.W.H., and J.T.S. conceived the project. C.B.S performed and oversaw all experiments with help from R.H., H.A., M.S., R.H., and J.M. D.B.R. performed all modeling, analysis, and generated all figures. M.F. and M.K.H. performed deuterium labeling experiments and analysis. P.W.H., S.G.D., J.M.M., and J.T.S. supervised the project. D.B.R. wrote the manuscript with many revisions from C.B.S., P.W.H., M.M., and J.T.S. All authors revised the manuscript.

## Competing interests

The authors declare no competing interests.

## Additional information

[1]Vaccine and Infectious Disease Division, Fred Hutchinson Cancer Center, 1100 Fairview Ave N, Seattle, WA 98109, USA. [2]Department of Global Health, University of Washington, 1959 NE Pacific St, Seattle, WA 98195, USA. [3]Vir Biotechnology, Inc, 1800 Owens Street Suite 900, San Francisco, CA 94158, USA. [4]Department of Nutritional Sciences and Toxicology, University of California, University Avenue and Oxford St, Berkeley, CA 94720, USA. [5]The Wistar Institute, 3601 Spruce St, Philadelphia, PA 19104, USA. [6]Department of Medicine, Zuckerberg San Francisco General Hospital, University of California, 1001 Potrero Ave, San Francisco, CA 94100, USA. [7]Division of Experimental Medicine, Department of Medicine, University of California San Francisco, 1001 Potrero Ave, San Francisco, CA 94100, USA. [8]Vitalant Research Institute, 360 Spear St Suite 200, San Francisco, CA 94105, USA. [9]Epidemiology & Biostatistics, University of California San Francisco School of Medicine, 550 16th Street, San Francisco, CA 94158, USA. [10]HIV Frontiers, Global Health Accelerator, Bill & Melinda Gates Foundation, 500 5th Ave N, Seattle, WA 98109, USA. [11]Clinical Research Division, Fred Hutchinson Cancer Center, 1100 Fairview Ave N, Seattle, WA 98109, USA. [12]Department of Allergy and Infectious Diseases, School of Medicine, University of Washington, 1959 NE Pacific St, Seattle, WA 98195, USA. [13]These authors contributed equally: Daniel B. Reeves, Charline Bacchus-Souffan, Joshua T. Schiffer, Peter W. Hunt. ✉e-mail: dreeves@fredhutch.org

