## [Peer Review file · Nature Communications]

nature portfolio

Peer Review FileREVIEWER COMMENTS

Reviewer #1 (Remarks to the Author):

This is a nice paper that offers new insights into the roles of the proliferation and differentiation of different resting CD4 T cell subsets in maintaining the HIV DNA reservoir. Rich data obtained from 37 people living with HIV on suppressive ART has been collected. Specifically, both cell proliferation and reservoir sizes in different cell subsets have been measured longitudinally. The data have been analysed using different mechanistic models and a mixed-effects approach to estimate the contribution of proliferation and differentiation to the maintenance of the HIV reservoir. The key findings are that in addition to proliferation, which previous studies have identified, differentiation can also contribute significantly to the reservoir. The contributions varied across subsets, which were quantified. The best model was then used to make predictions of the implications of combining ART with agents that may inhibit proliferation or increase differentiation to reduce the reservoir. Overall, the rich dataset together with its careful analysis renders the paper valuable. Some of the mechanistic insights, in my opinion, need further evaluation. The paper is well written. I have a few concerns and comments, which I list below.

1. While I understand the reduction in the reservoir upon inhibiting proliferation, I am unable to understand how enhancing differentiation would do the same. Differentiation would simply move cells from one compartment to another. Of course, the cells finally exit the differentiation cascade. I wonder if the exiting cells are counted in the reservoir. What is their fate? If they retain the HIV DNA, would the reservoir size decrease with increasing differentiation rates if they are counted in the reservoir? Alternatively, is it that increasing differentiation causes cells to focus into compartments with higher death rates? A clearer explanation of the implications of altering differentiation on the reservoir size would help.
2. The best model includes jumps in differentiation across compartments as opposed to a linear differentiation pathway through the compartments. I wonder if this is because of the sparseness in the data. For each patient, only 3 longitudinal measurements are available. Could such data naturally favour 3 compartments/transitions? It may help to ascertain this using simulated data. One could assume a linear pathway and generate data by solving the corresponding equations, but then mimic the current datasets using limited sampling. Would fitting as done here recover the linear pathway or favour the jumps?
3. Many mathematical models aim to predict remission following ART/other interventions that may help in devising cure strategies. It would help if the authors made explicit how their findings would inform such modelling or strategies. In particular, what is the role/importance of the various CD4 subsets in such models? Should they all be incorporated? How would that be advantageous over a parsimonious

description of a 1-2 compartment/subset model that accounts for proliferation and differentiation in a lumped manner?

Reviewer #2 (Remarks to the Author):

In the manuscript entitled “Estimating the contribution of CD4 T cell proliferation and differentiation to HIV persistence” by Reeves et al., the Authors measured, in 37 individuals with HIV-1 on suppressive antiretroviral therapy (ART), the cellular turnover of five CD4 T cell subsets: naïve (TN), stem-cell memory (TSCM), central memory (TCM), transitional memory (TTM), and effector memory (TEM) and measured the changes in integrated HIV DNA levels within these subsets over three years. Moreover, the Authors integrated these longitudinal data to a previous 45-day study of deuterium labeling obtained from 24 individuals with HIV-1 to measure CD4 T cell turnover rates using mathematical models. From this analysis the Authors observed that a) HIV clearance occurs mainly in TTM and TEM; b) CD4+ T cell turnover rates can be up to 10 times faster than HIV clearance rates; c) Maintenance of the cellular reservoir must be the most relevant factor to maintain HIV persistence. Finally, the Authors tested a number of mathematical models to estimate the levels of HIV infection and persistence by simulating different rates of cell proliferation and differentiation. The Authors conclude that CD4 T cell proliferation and, to a minor extent, differentiation have a relevant role in HIV persistence in the absence of viral replication.

Comments

1. The manuscript is a follow up study of the previous work by Bacchus-Souffan et al., published in PLOS Pathogens in 2021 (<https://doi.org/10.1371/journal.ppat.1009214>). Here the integrated HIV DNA levels of the same 37 individuals with HIV-1 studied previously were measured for 3 years (in about 3 time points for each individual). Besides the mathematical models tested in this work (all of which have some limitations as declared by the Authors), the main conclusions of the study are more or less the same of the previous publication. It is certainly nice to see that the conclusions in this longitudinal study are very similar to the previous one, but the advancement in the field is relatively small.

2. Moreover, it is unclear if the levels of integrated HIV-1 DNA were performed in this study or are derived from the previous study (and here the authors performed a stratification over the 3 years). I am asking this because there is no description of how the integrated HIV-1 DNA levels were measured. If the

Authors experimentally determined the integrated HIV-1 DNA levels in this work, they should specify that, tell what technique used in the results and materials an methods sections.

3. The experimental longitudinal data generated in is very superficial as there are only the levels of integrated HIV-1 DNA data reported for the 3 years follow up. There are many other data that can and should be generated to better support Authors' claims and to increase the novelty of the manuscript. These include: i) clonal tracking by integration site analysis to determine the clonality of the different T cell subsets and the type of targeted gens to confirm or exclude insertional mutagenesis and the proliferation levels of teach clone; ii) HIV-1 RNA measurements; iii) determination of the rates of defective and non-defective HIV-1 genomes.

Reviewer #3 (Remarks to the Author):

Defining host parameters leading to the persistence/maintenance of HIV reservoir is one of the most important challenge in the field of HIV cure. For this purpose, the authors used The HOPE cohort which consists of 37 people leaving with HIV on suppressive antiretroviral therapy (ART), 24 of whom received deuterium for period of 45 days for DNA labeling (ref). In this study, they extend their previous work by quantifying integrated HIV-1 DNA in five CD4 T cell subsets over period of 3 years (1-3 time points per participant) and measured CD4 T cell subsets turnover rate. The experiments are interesting and support the conclusions, although as stated by the authors, considerable variability was observed. Retrospectively, it is frustrating that the design of the experiments did not include methods allowing to distinguish replication competent reservoir from the reservoir containing defective virus and to analyze the impact of cell proliferation and differentiation on their maintenance. Regarding the mathematical modeling, it ignores the impact of HIV on cell proliferation, survival and differentiation. Indeed, recent study revealed that HIV persistent CD4 T cells show phenotypic signature primarily characterized by upregulation of surface markers promoting cell survival (Sun W. et al. Nature. 2023 Feb;614(7947):309-317 ; Clark IC et al. Nature. 2023 Feb;614(7947):318-325). If possible, the authors can include this information in their modeling or at least discuss these recent findings accordingly.

REVIEWER COMMENTS

Reviewer #1 (Remarks to the Author):

This is a nice paper that offers new insights into the roles of the proliferation and differentiation of different resting CD4 T cell subsets in maintaining the HIV DNA reservoir. Rich data obtained from 37 people living with HIV on suppressive ART has been collected. Specifically, both cell proliferation and reservoir sizes in different cell subsets have been measured longitudinally. The data have been analysed using different mechanistic models and a mixed-effects approach to estimate the contribution of proliferation and differentiation to the maintenance of the HIV reservoir. The key findings are that in addition to proliferation, which previous studies have identified, differentiation can also contribute significantly to the reservoir. The contributions varied across subsets, which were quantified. The best model was then used to make predictions of the implications of combining ART with agents that may inhibit proliferation or increase differentiation to reduce the reservoir. Overall, the rich dataset together with its careful analysis renders the paper valuable. Some of the mechanistic insights, in my opinion, need further evaluation. The paper is well written. I have a few concerns and comments, which I list below.

We would like to thank the reviewer sincerely for this thoughtful reading and review.

1. While I understand the reduction in the reservoir upon inhibiting proliferation, I am unable to understand how enhancing differentiation would do the same. Differentiation would simply move cells from one compartment to another. Of course, the cells finally exit the differentiation cascade. I wonder if the exiting cells are counted in the reservoir. What is their fate? If they retain the HIV DNA, would the reservoir size decrease with increasing differentiation rates if they are counted in the reservoir? Alternatively, is it that increasing differentiation causes cells to focus into compartments with higher death rates? A clearer explanation of the implications of altering differentiation on the reservoir size would help.

Thanks for this clarification, we agree with this reviewer's phrasing: "increasing differentiation causes cells to focus into compartments with higher death rates". Moreover, the modeling of measured HIV DNA levels within each CD4 T cell subset over time demonstrated that TEM cells carrying integrated HIV DNA decay more rapidly than those in less mature CD4 subpopulations. While the precise mechanism for this more rapid clearance of HIV DNA from TEM CD4 cells is not currently understood, promoting differentiation would push cells towards this maturation stage and out of generally slower clearing subsets. We have now explicitly made this point in the discussion.

2. The best model includes jumps in differentiation across compartments as opposed to a linear differentiation pathway through the compartments. I wonder if this is because of the sparseness in the data. For each patient, only 3 longitudinal measurements are available. Could such data naturally favour 3 compartments/transitions? It may help to ascertain this using simulated data. One could assume a linear pathway and generate data by solving the corresponding equations, but then mimic the current datasets using limited sampling. Would fitting as done here recover the linear pathway or favour the jumps?

We thank the reviewer for this very creative idea. We have now performed a sensitivity analysis on the model selection scheme as suggested here. We generated data from the linear differentiation model, added noise, and sampled per the experimental protocol. Then, we used our model selection process to determine the optimal model for these

simulated data. The result was that by likelihood and Akaike Information Criterion (AIC), the same model generating the data (linear differentiation) was also the model selected to best fit those simulated data. Thus, it shows the model selection in this case is self-consistent. We are grateful for this idea because we think it supports our model selection procedure and the optimal model. It also suggests that the data/noise/sampling scheme do not intrinsically favor the skip-differentiation model. This analysis is now included as Supplementary Figure 7 and in a paragraph of the results.

3. Many mathematical models aim to predict remission following ART/other interventions that may help in devising cure strategies. It would help if the authors made explicit how their findings would inform such modelling or strategies. In particular, what is the role/importance of the various CD4 subsets in such models? Should they all be incorporated? How would that be advantageous over a parsimonious description of a 1-2 compartment/subset model that accounts for proliferation and differentiation in a lumped manner?

We thank the reviewer for this idea. Although we considered a simpler model with only 2 compartments (naïve and combined memory), we ultimately decided not to show this in the current manuscript because it would obscure our important biological findings of HIV DNA decaying in certain subsets (TTM and TEM) vs TCM.

However, inspired by this excellent suggestion, we provide a new Supplementary Table 2 of all initial conditions and population parameters (and their variances) that would help future modelers to simulate the full model in more generalized scenarios and allow others to make simplified/combined projections if desired.

Reviewer #2 (Remarks to the Author):

In the manuscript entitled “Estimating the contribution of CD4 T cell proliferation and differentiation to HIV persistence” by Reeves et al., the Authors measured, in 37 individuals with HIV-1 on suppressive antiretroviral therapy (ART), the cellular turnover of five CD4 T cell subsets: naïve (TN), stem-cell memory (TSCM), central memory (TCM), transitional memory (TTM), and effector memory (TEM) and measured the changes in integrated HIV DNA levels within these subsets over three years. Moreover, the Authors integrated these longitudinal data to a previous 45-day study of deuterium labeling obtained from 24 individuals with HIV-1 to measure CD4 T cell turnover rates using mathematical models. From this analysis the Authors observed that a) HIV clearance occurs mainly in TTM and TEM; b) CD4+ T cell turnover rates can be up to 10 times faster than HIV clearance rates; c) Maintenance of the cellular reservoir must be the most relevant factor to maintain HIV persistence. Finally, the Authors tested a number of mathematical models to estimate the levels of HIV infection and persistence by simulating different rates of cell proliferation and differentiation. The Authors conclude that CD4 T cell proliferation and, to a minor extent, differentiation have a relevant role in HIV persistence in the absence of viral replication.

Comments

1. The manuscript is a follow up study of the previous work by Bacchus-Souffan et al., published in PLOS Pathogens in 2021 (<https://doi.org/10.1371/journal.ppat.1009214>). Here the integrated HIV DNA levels of the same 37 individuals with HIV-1 studied previously were measured for 3 years (in about 3 time points for each individual). Besides the mathematical models tested in this work (all of which have some limitations as declared by the Authors), the main conclusions of the study are more or less the same of the previous publication. It is certainly nice to see that

the conclusions in this longitudinal study are very similar to the previous one, but the advancement in the field is relatively small.

We appreciate the reviewer's reading and familiarity with our previous manuscript, which was focused on a subgroup of 24 study participants solely at baseline. As this reviewer highlighted, the current manuscript includes the whole HOPE cohort that consisted of 37 participants, who were followed annually for 3 consecutive years.

Although we were glad to see results that were consistent with our team's previous findings, there are new data reported in the current manuscript that led to several important novel findings:

- 1) By directly measuring HIV DNA clearance rates within T cell subsets over 3 years, we directly compare (for the first time) CD4 T cell turnover to HIV turnover within the same individual(s). This analysis led to the novel and potentially important observation that HIV is cleared more rapidly in more differentiated subsets (TEM).**
- 2) Modeling the longitudinal data substantially enhances our quantitative understanding of mechanistic biology underlying HIV persistence. Previously, we observed that differentiation appeared to be detectable, but could not estimate the relative contribution of cell proliferation relative to differentiation to HIV persistence.**
- 3) Finally, the mathematical model we derive here provides potentially the most detailed and data driven approach for exploratory simulations of HIV DNA dynamics during ART + additional curative therapies. Reviewer 1 suggested we emphasize this point more, which is now explored further in the discussion.**

2. Moreover, it is unclear if the levels of integrated HIV-1 DNA were performed in this study or are derived from the previous study (and here the authors performed a stratification over the 3 years). I am asking this because there is no description of how the integrated HIV-1 DNA levels were measured. If the Authors experimentally determined the integrated HIV-1 DNA levels in this work, they should specify that, tell what technique used in the results and materials and methods sections.

We apologize for this confusing oversight, we have now included methods sections for cell sorting and viral quantifications, which were performed similarly to the prior work for consistency.

Specifically for the current manuscript, we collected baseline levels of integrated HIV DNA for 13 additional participants beyond the original 2021 article, as well as at the novel year 2 and year 3 time points for most of the 37 participants.

3. The experimental longitudinal data generated in is very superficial as there are only the levels of integrated HIV-1 DNA data reported for the 3 years follow up. There are many other data that can and should be generated to better support Authors' claims and to increase the novelty of the manuscript. These include: i) clonal tracking by integration site analysis to determine the clonality of the different T cell subsets and the type of targeted gens to confirm or exclude insertional mutagenesis and the proliferation levels of each clone; ii) HIV-1 RNA measurements; iii) determination of the rates of defective and non-defective HIV-1 genomes.

We thank the reviewer for these excellent suggestions. We did perform some of the analyses suggested, and we explain below why some of the others are unfortunately not possible for these samples:

i) The longitudinal integration sites analysis was performed in collaboration with the Lewin lab but was unfortunately lost due to technical hurdles, so this data will regretfully not be available on this dataset.

ii) We now include a parallel analysis on cell-associated HIV RNA in Supplementary Figure 8, and a paragraph in the results.

iii) Intact/defective provirus quantifications were performed on a subset of participants in two published manuscripts (Morcilla 2021 mBio; Duette-Hiener 2022 JCI). However, none of these findings were on longitudinal samples since they assessed at baseline (year 1). Nonetheless, looking at some of these data (Table 3 from Morcilla et al. mBio copied below), we unfortunately suspect that modeling separated by subsets would be practically impossible due to very few measured intact sequences found in each subset (mostly 0 or 1 copy found if even available). For these reasons, modeling integrated HIV DNA is the only approach possible at present.

[redacted]

Reviewer #3 (Remarks to the Author):

Defining host parameters leading to the persistence/maintenance of HIV reservoir is one of the most important challenge in the field of HIV cure. For this purpose, the authors used The HOPE cohort which consists of 37 people leaving with HIV on suppressive antiretroviral therapy (ART), 24 of whom received deuterium for period of 45 days for DNA labeling (ref). In this study, they extend their previous work by quantifying integrated HIV-1 DNA in five CD4 T cell subsets over period of 3 years (1-3 time points per participant) and measured CD4 T cell subsets turnover rate. The experiments are interesting and support the conclusions, although as stated by the authors, considerable variability was observed. Retrospectively, it is frustrating that the design of the experiments did not include methods allowing to distinguish replication competent reservoir from the reservoir containing defective virus and to analyze the impact of cell proliferation and differentiation on their maintenance. Regarding the mathematical modeling, it ignores the impact of HIV on cell proliferation, survival and differentiation. Indeed, recent study revealed that HIV persistent CD4 T cells show phenotypic signature primarily characterized by upregulation of surface markers promoting cell survival (Sun W. et al. Nature. 2023 Feb;614(7947):309-317; Clark IC et al. Nature. 2023 Feb;614(7947):318-325). If possible, the authors can include this information in their modeling or at least discuss these recent findings accordingly.

We thank the reviewer for their fair reading and understand their frustration. It would have been highly relevant to have modeled intact vs. defective genomes individually, and we are left wondering whether we would have seen signals of selectively more blocked/locked proviruses over time in the intact pool. Although some intact/defective analyses were performed and published on baseline samples for a subset of study participants (see response above to Reviewer 2), the numbers of intact sequences were simply too low to even perform basic correlation analyses much less detailed differential equations based modeling, and we did not feel this could support the longitudinal collection to attempt modeling.

We highly appreciate your point about the impact of HIV on cell fate and have emphasized two key caveats you brought up in our revised discussion. First, large clones, which may also be the most proliferative, may also be less likely to reactivate if ART is stopped. Together this means a careful interpretation of anti-proliferative therapy is required. Although it was tempting to make a model of the relationship between proliferative capacity and rebound potential, we felt it was not yet appropriate without any data on how HIV's effect on cell survival (or proliferation/differentiation) may vary by CD4 subset.

Instead, we have acknowledged more fully the assumption in our model is that turnover of resting HIV-infected CD4 cells is assumed to be comparable to the turnover of resting HIV-uninfected cells, which may not be the case. We have also referenced the papers cited by the reviewer as examples of data suggesting that HIV-infected cells may have altered cell survival pathways compared to uninfected cells. On the other hand, HIV-infected cells may also be more susceptible to immune-based clearance than uninfected cells. While the precise contribution of each of these pathways remains unknown in vivo, the sum total of the forces suggests that HIV-infected cells are cleared more rapidly than HIV-uninfected cells in more differentiated T cell subsets.

REVIEWERS' COMMENTS

Reviewer #1 (Remarks to the Author):

The authors have done a good job of addressing my comments. I have no further comments.

Reviewer #2 (Remarks to the Author):

The Authors addressed satisfactorily this reviewer's concerns

Reviewer #3 (Remarks to the Author):

Happy with the answers and changes made.